# Spatial Correlation and Influencing Factors of Tourism Eco-Efficiency in the Urban Agglomeration of the Yangtze River Delta Based on Social Network Analysis

**Yuewei Wang [1], Lidan An [1], Hang Chen [2],\* and Yuyan Zhao [3]**

[1]   School of Business, Liaoning University, Shenyang 110036, China
[2]   School of Tourism Management, Shenyang Normal University, Shenyang 110034, China
[3]   School of Management, Shenyang Jianzhu University, Shenyang 110168, China
\*   Correspondence: chenhang-119@163.com

**Abstract:** Tourism eco-efficiency analysis is an effective tool to solve the problem of sustainable tourism development. The tourism eco-efficiency evaluation index system was constructed in the study, and the undesirable output super-slacks-based measure model was used to estimate the tourism eco-efficiency of 26 cities in the Yangtze River Delta. Then, the modified gravity model based on the values of the tourism eco-efficiency analysis of each city was used to construct a spatial correlation network. The structural characteristics of the spatial association networks of tourism eco-efficiency, the interrelationships among different cities, and the roles played by different blocks were explored using a social network analysis. The quadratic assignment procedure (QAP) was applied to analyze the influencing factors that affect the formation of the spatial association network of tourism eco-efficiency. The results show that tourism eco-efficiency has an overall increasing trend, and the gap among cities is decreasing. The structure of the spatial correlation network of tourism eco-efficiency has good connectivity, accessibility, and robustness with the correlations among all of the cities in the network. The spillover effects among the blocks are significant, showing spatial polarization, with the cities such as Shanghai, Suzhou, and Hangzhou occupying the core position of the network. The QAP analysis shows that the spatial correlation network of tourism eco-efficiency is affected by the distance between the cities and the levels of development of the economy and information dissemination. The results of this study innovatively reveal the structural characteristics and influencing factors of the spatial correlation network of tourism eco-efficiency. It could provide valuable insights for the development of corresponding policy measures by government sectors and tourism firms to enhance the sustainability of regional tourism development.

**Keywords:** undesirable output super-slacks-based measure model; spatiotemporal effects; quadratic assignment procedure; block models

## 1. Introduction

Tourism eco-efficiency has a profound impact on the sustainable development of global tourism. The scale of human tourism activities is continuously increasing. The growth of tourism has significantly benefitted the tourism-receiving communities, economically, but it has also had many detrimental repercussions on the local environment, community, and culture. The negative effects of tourism on the environment have obvious cascading effects, making it difficult to remedy the "pollution first, then treatment" problem. This is due to the unique geographical conditions of the tourism areas and the relative concentration of tourist flow over time (tourist season) and space (tourist hotspots), resulting in a cycle of pollution that is caused by tourists. Therefore, the communities must seriously consider ways to make tourism sustainable over the long term to reap long-term economic and social advantages. Tourism depends on the environment for its existence and development, but it may also harm or even destroy it. Eco-efficiency seeks to strike a balance between environmental

protection and economic gains, and its goal is to reduce the impact of the economic benefits on the environment to realize the harmonious development of local economic activities and ecological environment. Usually, the ratio of output-to-input is used to express eco-efficiency [1]. The value of the goods and services produced by a business or economic unit is referred to as its "output", and the consumption of its resource and energy and imposed environmental burden are referred to as its "input" [1]. An essential sign of sustainable tourism development is tourism eco-efficiency, which functions "on the basis of ensuring minimizing resource input and environmental damage and maximizing socioeconomic benefits as much as possible" [1]. The impact of the development of tourism in the current and long-term perspectives on the ecological environment [2] is based on the harmonious and balanced development of tourism resources and ecological environment [3] with the improvement of the quality of tourism development. In addition, the joint promotion of the regional linkage development strategy and the market mechanism has strengthened the spatial connection of tourism eco-efficiency. With the development of regional coordination and integration, the spatial relationship of tourism eco-efficiency presents a more complex network structure. Therefore, it is indispensable to further investigate tourism eco-efficiency and compare its spatiotemporal differences, spatial correlations, and influencing factors in different cities, and these are particularly essential for exploring the environmentally friendly and balanced development. The investigation of tourism eco-efficiency can provide new ideas for environmentally friendly eco-tourism and the sustainable development of urban industry. Tourism eco-efficiency plays a vital part in accelerating urban construction at the economic level, transforming and upgrading at the industrial level, and improving the development level of an urban agglomeration.

With the accelerated development of global tourism and the increased impacts of tourism activities on the environment, the eco-benefits of tourism are beginning to receive attention. For example, Gossling et al. analyzed the eco-efficiency of tourism in Seychelles and Amsterdam based on the carbon dioxide-equivalent emissions [4]. Kelly et al. explored the issue of optimizing eco-efficient tourism planning in tourist destinations from the tourist's perspective [5]. Becken used ten eco-efficiency indicators to measure the dependence of New Zealand's top ten international source markets on oil-based energy consumption [6]. Patterson et al. used the tourism satellite account and input–output tables to study the eco-efficiency of tourism in New Zealand [7]. The evaluation of tourism eco-efficiency usually includes both the economic income and the eco-environmental impact indicators [8–10]. Evaluations of tourism eco-efficiency have been performed more often on restaurants and accommodation businesses. For example, Li et al. used the bootstrap data envelopment analysis (DEA) to measure and calculate the eco-efficiency of the hotel industry in 31 provinces of China from 2016 to 2019 [11]. Xia et al. analyzed the spatial patterns of inputs and outputs and development trends of the eco-economic system of star-rated hotels in China based on the calculated levels of their economic efficiency and eco-efficiency in 30 provinces in China [12]. In previous studies, scholars have mainly applied the indicator system, a model analysis, and the economic–ecological single ratio methods to evaluate tourism eco-efficiency. For example, the measurement of tourism eco-efficiency in the economic–ecological single ratio method is expressed as the ratio of tourism revenue-to-environmental indicators such as the carbon emissions [3,13–15]. Recently, scholars have started to use more complex models to evaluate tourism eco-efficiency. These models mainly include the input–output methods [3], the DEA, and the stochastic frontier analysis [16]; of these, the DEA models are more widely used. For example, Li et al. used the super-slacks-based measure (SBM) DEA (SBM-DEA) with an undesirable output model to calculate the eco-efficiency values of provincial tourism in China [17]. Wang et al. constructed multiple regional tourism input–output index systems and calculated the tourism input–output ratios of 31 provinces in China from 1997 to 2016 using the undesirable output model in a DEA. Li et al. analyzed the spatial autocorrelation of the ecological efficiency of 28 national forest parks using the undesirable output super-SBM model of a DEA [18,19].

In general, the previous studies on tourism eco-efficiency have mainly focused on concept definition [20], model construction [5], efficiency measurement [4,21], and the influencing factors, and the formation mechanism [2,5]. However, previous studies have not addressed two aspects. Firstly, the previous studies were performed at the national [22], provincial [23,24], watershed [25], prefecture-level city [26,27], and tourist scenic area [28] scales. In these studies, tourism eco-efficiency was largely assessed based on individual spaces even though some of them confirm that the tourism eco-efficiencies of different regions are interdependent with some spatial correlation [29,30]. In addition, most of the existing literature adopts traditional spatial measures. These studies often limit the spatial correlation to geographically adjacent or similar regions [17–19,31], and they fail to explore the structural characteristics and driving factors of the spatial correlation of tourism eco-efficiency as a whole. Secondly, most of the existing literature is based on an "attribute data" analysis [16], and they can reflect only on the current situation of tourism eco-efficiency in each region without revealing the structural characteristics of the spatial linkages of the network of tourism eco-efficiency. Some studies have analyzed the structural characteristics of the spatial linkages of tourism eco-efficiency of the complete network from the perspective of "relationships". A social network analysis is an interdisciplinary and cross-domain research method based on relational data, and it is a new research paradigm in the fields of sociology, management, and economics [32,33]. However, few scholars have applied a social network analysis to study the spatial correlation of tourism eco-efficiency. This study uses the undesirable output super-SBM model to calculate the tourism eco-efficiency of 26 cities in the Yangtze River Delta region by constructing the tourism eco-efficiency index system. The spatial correlation network of eco-efficiency of inter-city tourism was constructed using the modified gravity model. Furthermore, the structural characteristics and influencing factors of the spatial correlation network of tourism eco-efficiency were analyzed using the social network analysis method.

The research questions that are posed in this study are as follows:

1. Are there spatial differences in the tourism eco-efficiency in different cities?
2. What is the spatial distribution pattern and evolutionary trend of tourism eco-efficiency?
3. What are the roles of different cities in the spatial correlation network of tourism eco-efficiency, and do they form different blocks?
4. What are the internal and external receiving or sending relationships of each block?
5. What factors, if any, have a significant impact on the formation of the spatial correlation network of tourism eco-efficiency?

From a practical point of view, it is very important to understand the spatial correlation of tourism eco-efficiency. The main suppliers of the cities with different tourism eco-efficiency values should include the government and tourism enterprises in the Yangtze River Delta. The government plays a leading role in the interaction of cities with different tourism eco-efficiency. Through the formulation of regional tourism cooperation plans, the government will make an overall allocation of capital, labor, energy, tourism resources, tourism services, and other aspects within the different cities. In response, tourism enterprises jointly design tourism routes, carry out publicity and promotional activities, provide reception services, and conduct other measures. This encourages the cities with different tourism eco-efficiencies to interact spatially. That is, the input behavior of the government and tourism enterprises forms the spatial correlation of tourism eco-efficiency. Therefore, the five questions above can provide valuable insights for the development of corresponding policy measures by government sectors and tourism firms to enhance the sustainability of regional tourism development. Four sub-goals were established to answer these questions. Firstly, the evaluation index system of tourism eco-efficiency was constructed from the aspects of input and output. The tourism inputs include five indicators of capital, labor, tourism resources, tourism services, and energy. The tourism outputs include the desirable and undesirable outputs. Secondly, the undesirable output super-SBM model was applied to measure the tourism eco-efficiency values of the different cities in the Yangtze River Delta, and their spatial differences and evolution patterns were analyzed. Thirdly, using

the modified gravity model, the value of tourism eco-efficiency was transformed into the spatial correlation network of inter-city tourism eco-efficiency. This study used the social network analysis method to explore the structural characteristics of the spatial correlation network of tourism eco-efficiency, the relationships between the different cities, and the roles of the different functional segments. Fourthly, the quadratic assignment procedure (QAP) was applied to analyze the influencing factors that can affect the formation of the spatial association network of tourism eco-efficiency.

## 2. Materials and Methods

### 2.1. Study Area

The Yangtze River Delta is the junction area of the Belt and Road Initiative and the economic belt along the Yangtze River Basin, which plays a vital role in China's national modernization and opening-up program. According to the "Yangtze River Delta Development Plan" released by the General Office of the State Council in 2016, the Yangtze River Delta urban agglomeration consists of Nanjing, Changzhou, Shanghai, Wuxi, Suzhou, Nantong, Yancheng, Yangzhou, Zhenjiang, and Taizhou in Jiangsu Province, Hangzhou, Jiaxing, Ningbo, Huzhou, Shaoxing, Jinhua, Zhoushan, and Taizhou in Zhejiang Province, and Hefei, Ma'anshan, Wuhu, Chuzhou, Tongling, Anqing, Chizhou, and Xuancheng in Anhui Province. These 26 cities are a constant source of economic growth for China. In 2019, the tourism industry in the Yangtze River Delta was in a good condition. In Shanghai, Zhejiang, Jiangsu, and Anhui, the total number of tourist visits and the total revenue of tourism increased rapidly. The four provinces and cities in the Yangtze River Delta region received more than 370 million tourists, and the total number of tourists reached 2.811 billion, which is going up by 8.43% year-on-year, accounting for 32.62% of the total number of tourists in China. The total tourist income was CNY 3.91 trillion, which is going up by 12.3% year-on-year, accounting for 36.63% of the total tourist income in China. Analyses of the spatiotemporal evolution characteristics and dynamical mechanisms of tourism eco-efficiencies of these cities can provide valuable inputs for the sustainable development of tourism in cities of other regions and countries.

### 2.2. Construction of Tourism Eco-Efficiency Index System

This study constructed the tourism eco-efficiency evaluation index system based on previous studies combined with the available data for the cities in the Yangtze River Delta using the input and output variables (Table 1).

**Table 1.** Tourism eco-efficiency evaluation index.

| Indicator Type | Indicator Name | Primary Parameter | References |
|---|---|---|---|
| Input | Capital input | Investment in fixed assets for tourism | Lu et al. [8]; Yang et al. [9]; Wang et al. [10]; Wang et al. [29]; Chaabouni [31]; Wang [34]; Hu [35]. |
| | Labor input | Number of tourism employees | Lu et al. [8]; Yang et al. [9]; Wang et al. [10]; Chaabouni [31]; Wang [34]; Hu [35]. |
| | Tourism resource | Number of A-level scenic spots | Lu et al. [8]; Yang et al. [9]; Wang et al. [29]; Wang [34]; Hu [35]. |
| | Tourism service | Number of star hotels | Yang et al. [9]; Wang et al. [29]; Wang [34]; Hu [35]. |
| | | Number of travel agencies | Wang et al. [29]; Wang [34]; Hu [35]. |
| | Energy input | Water supply | Peng et al. [27]; Guo et al. [36]. |
| | | Power supply | Peng et al. [27]; Guo et al. [36]. |
| Desirable Output | Total tourism economy | Total tourism revenue | Lu et al. [8]; Yang et al. [9]; Wang et al. [10]; Chaabouni [31]; Wang [34]; Hu [35]. |
| | | Visitor reception volume | Lu et al. [8]; Yang et al. [9]; Wang et al. [10]; Chaabouni [31]; Wang [34]. |
| Undesirable Output | Tourism environmental pollution | Wastewater discharge from tourism | Lu et al. [8]; Yang et al. [9]; Wang et al. [10]. |
| | | $SO_2$ emissions from tourism | Lu et al. [8]; Yang et al. [9]; Wang et al. [10]. |
| | | Smoke emissions from tourism | Yang et al. [9]. |

*2.3. Methods*

2.3.1. Undesirable Output Super-SBM Model

The undesirable output super-SBM model proposed by Tone is an improvement on the conventional DEA model as it corrects the bias developed because of the radial and directional problems and accurately assesses the relationship between the inputs and outputs [37]. This study used this model to gauge the tourism eco-efficiency [37]. The formulas used are as follows:

$$\rho = \min \frac{1 + \frac{1}{m}\Sigma_{i=1}^{m}\frac{s_i^x}{x_{i0}}}{1 - \frac{1}{s_1+s_2}\left(\Sigma_{k=1}^{s_1}\frac{s_k^y}{y_{k0}} + \Sigma_{l=1}^{s_2}\frac{s_l^z}{z_{l0}}\right)} \tag{1}$$

$$\text{s.t. } x_{i0} \geq \Sigma_{j=1,\neq 0}^{n}\lambda_j x_j - s_i^x, \forall i \tag{2}$$

$$y_{k0} \leq \Sigma_{j=1,\neq 0}^{n}\lambda_j y_j - s_k^y, \forall k \tag{3}$$

$$z_{l0} \leq \Sigma_{j=1,\neq 0}^{n}\lambda_j z_j - s_l^z, \forall l \tag{4}$$

$$1 - \frac{1}{s_1+s_2}\left(\Sigma_{k=1}^{s_1}\frac{s_k^y}{y_{k0}} + \Sigma_{l=1}^{s_2}\frac{s_l^z}{z_{l0}}\right) > 0 \tag{5}$$

$$s_i^x \geq 0, s_j^y \geq 0, s_l^z \geq 0, \lambda_j \geq 0, \forall i, j, k, l \tag{6}$$

where $\rho$ indicates the tourism eco-efficiency value; m, $s_1$, and $s_2$ represent the number of variables of input, the desirable output, and the undesirable output, respectively; $x_{i0}$, $y_{k0}$, and $z_{l0}$ are the elements in the corresponding input, desirable output, and undesirable output matrices, respectively; $\lambda$ is the weight vector. Here, $s^x$, $s^y$, and $s^z$ are not slack variables in the traditional sense, but they are slack variables indicating the inputs, the desirable outputs, and the undesirable outputs, respectively.

2.3.2. Modification of the Gravity Model

Gravity is a law of nature that was originally published by Isaac Newton in 1687 in his work 'Mathematical Principles of Natural Philosophy'. The law of gravity was later applied to the field of economics. Economists believe that there is a law of economic linkage between regions which is similar to gravity, i.e., the strength of the linkage between regions is proportional to the "mass" of the regions; the strength of the linkage between regions is inversely proportional to the "distance" between the regions, as influenced by the law of decay of distance [38]. Since Zipf first applied the modified gravitational model to the analysis of inter-city connections in 1942, the gravitational model has been widely used in the study of regional spatial interconnection [39–41]. Therefore, the interrelationships of tourism eco-efficiency of the different cities in Yangtze River Delta can be investigated using the modified gravitational model according to the following equations:

$$F_{ij} = K_{ij}\frac{E_i \cdot E_j}{D_{ij}^2} \tag{7}$$

$$K_{ij} = \frac{E_i}{E_i + E_j} \tag{8}$$

$$D_{ij}^2 = \left(\frac{d_{ij}}{g_i - g_j}\right)^2 \tag{9}$$

where *Fij* is the intensity of the tourism eco-efficiency linkage of each city; *Kij* is the gravitational coefficient; *Ei* and *Ej* represent the tourism eco-efficiency of city *i* and city *j*, respectively; *Dij* indicates the "economic distance" between city *i* and city *j*; *dij* is the spherical distance between the cities; *gi* and *gj* represent the GDPs of city *i* and city *j*, respectively [40]. The

spatial correlation matrix of tourism eco-efficiency is structured by the gravity model, and the mean value of each row of data in the matrix is used as the threshold for then binarization. If *Fij* is greater than the mean value, then it takes the value of 1, which means that there is a spatial correlation between the cities in terms of tourism eco-efficiency, and vice versa, if it takes the value of 0, that means that there is no spatial correlation.

### 2.3.3. Social Network Analysis

Based on the gravity model, this study calculated the spatial correlation of tourism eco-efficiency in the Yangtze River Delta urban agglomeration and constructed the relationship matrix, which was further binarized. Then, Ucinet 6.0 software was used to analyze the social network of the processed matrix.

(1)    Characteristics of the whole network

The overall network characteristics are mainly measured by their density, connectedness, hierarchy, and efficiency, reflecting the spatial association network's strength of association and structure of tourism eco-efficiency (Table 2). The network density is expressed as the ratio of the actual number of tourism eco-efficiency relationships between the cities in the network to the theoretical maximum number of relationships. Network connectedness measures the degree of connectedness in the tourism eco-efficiency spatial network. The network hierarchy measures the quantum of asymmetric accessibility that exists between the cities in the tourism eco-efficiency spatial network. Network efficiency measures the efficiency of connectivity between the cities in the tourism eco-efficiency spatial network.

**Table 2.** Formulas for the structural characteristics of spatial correlation network of tourism eco-efficiency.

| Indicator Type | Indicator Name | Formula | Meaning of Variables |
|---|---|---|---|
| Characteristics of the whole network | Network density (*ND*) | $ND = a/[b(b-1)/2]$ | $a$ is the number of cities |
| | Network connectedness (*NC*) | $NC = 1 - V/[b(b-1)/2]$ | $V$ is the number of "0s" on the diagonal of the reachable matrix |
| | Network hierarchy (*NH*) | $NH = 1 - S/max(S)$ | $S$ stands for the logarithms of symmetric nodes in spatially associated networks |
| | Network efficiency (*NE*) | $NE = 1 - K/max(K)$ | $K$ is the actual number of redundant relationships in the spatial association network structure |
| Characteristics of the individual network | Degree of centrality (*DC*) | $DC = d/(b-1)$ | $d$ is the number of direct relationships between a city and other cities in a spatial association network |
| | Closeness centrality (*CC*) | $CC = \sum_{j=1}^{b} l_{ij}$ | $l_{ij}$ distance, namely in the shortcut contains the number of relations |
| | Betweenness centrality (*BC*) | $BC = \frac{2\sum_{p}^{b}\sum_{q}^{b} n_{pq}(i)}{b^2 - 3b + 2}$ $n_{pq}(i) = g_{pq}(i)/g_{pq}$ | $g_{pq}$ is the number of the shortcuts between the node $p$ and the node $q$; $g_{pq}(i)$ is the number of the node $p$ and node $q$ after the node $i$; $n_{pq}(i)$ means the probability that node $i$ is in the number of shortcuts between nodes $p$ and $q$; $p \neq q \neq i$, $p < q$ |

(2)    Characteristics of the individual network

The characteristics of an individual network mainly describe the position and role of each city in the spatial correlation network of tourism eco-efficiency through centrality, and the measurement indicators are the degree, closeness, and betweenness of the centrality. The degree of centrality measures the centrality of each city in the spatial correlation network of tourism eco-efficiency according to the number of connections. Closeness centrality is

the degree to which a city is directly related to other cities in the spatial network of tourism eco-efficiency. Betweenness centrality measures the extent to which a city is in the "middle" of the tourism eco-efficiency transmission path of other cities.

2.3.4. Quadratic Assignment Procedure (QAP)

In this study, a QAP analysis was applied to analyze the driving factors of the spatial correlation network of tourism eco-efficiency. The QAP is an arrangement based on the matrix data, and it can compare the similarity of individual cells in two or more matrices. That is, the single cell of the matrix is compared, and the correlation coefficient between the two matrices is given. Then, the coefficient is non-parametrically tested. The individual observations of the relational data are not independent of each other. Parameter estimation and statistical tests cannot be performed with many standard statistical procedures (e.g., ordinary least squares regression) because the observation terms are not independent of each other, and incorrect standard deviations are calculated. Therefore, scholars use a randomization test (e.g., randomly controlled trial) method that is termed QAP, which is a resampling-based method that excludes the spurious structural relationships. The QAP is commonly applied in tourism, sociology, and geography studies [38,39]. The QAP can analyze the driving factors of tourism eco-efficiency development based on no pseudo-structure relationship, therefore, it better reflects the interrelationships between the cities compared to other regression analysis methods, and it provides more realistic, reliable, and specific insights for improvement, thereby enabling the sustainable development of tourism in the long run.

This study constructs the evaluation model of driving factors of the spatial correlation network of tourism eco-efficiency:

$$R = f\,(EDL, TIS, TIL, TRE, DBC, IDL) \tag{10}$$

where R is the tourism eco-efficiency network relationship matrix, Economic development level (EDL) denotes the difference matrices of the Economic development level, Tourism investment level (TIL) denotes the difference matrices of the tourism industry status, Tourism investment level (TIL) denotes the difference matrices of the Tourism investment level, Tourism resource endowment (TRE) denotes the difference matrices of the Tourism resource endowment, Distance between cities (DBC) denotes the difference matrices of the Distance between cities, and Information development level (IDL) is the difference matrices of the Information development level.

EDL: The EDL of a city has a vital contribution to the development of tourism ecological resources, the construction of infrastructure, and the betterment of the ecological environment in which the tourism occurs. Cities with a high EDL invest more in tourism development with relatively well-developed attractions and infrastructure and a relatively efficient flow of tourism production resources, indicating the good connectivity of tourism eco-efficiency between the cities. Therefore, the EDL is a vital factor that affects the spatial correlation network.

TIS: The TIS reflects the government's intention and goal to achieve regional economic development based on tourism. The more the government pays attention to the development of the tourism industry, the more importance it attaches to it, and the more helpful it is to improve the eco-efficiency of tourism. Therefore, the TIS will affect the spatial correlation network.

TIL: The higher the capital injection is, the more guarantee that there will be a quality of the tourism elements. The TIL also promotes more frequent inter-city communication in the tourism industry, thus promoting inter-city cooperation in tourism production, and finally, it has an impact on the spatial correlation network.

TRE: The TRE can be used as the primary attraction of the development of the tourism industry in each city. Therefore, the TRE is an important factor that affects the spatial correlation network.

DBC: The farther the DBC is, the more it will affect the transportation and information transfer efficiencies of the tourism elements between the cities, which in turn will affect the spatial correlation network.

IDL: The higher the IDL is, then the more convenient the flow of the tourism production factors are. Therefore, it is an influencing factor of the spatial correlation network.

## 3. Results and Discussion

### 3.1. Spatial Differences in Tourism Eco-Efficiency in Different Cities

There are spatial differences in tourism eco-efficiency in different cities. The natural breaks (Jenks) method in ArcGIS 10.5 software was used to grade data in 2009 and 2019. The data were classified into grades I, II, III, IV, and V according to the color from dark to light (Figure 1). Through a comparative analysis, the spatial differentiation and dynamic change patterns of tourism eco-efficiency were revealed. In previous studies, some scholars have adopted similar classification methods, but they generally focus on the spatial differences of urban tourism eco-efficiency in countries and provinces [18,42–45].

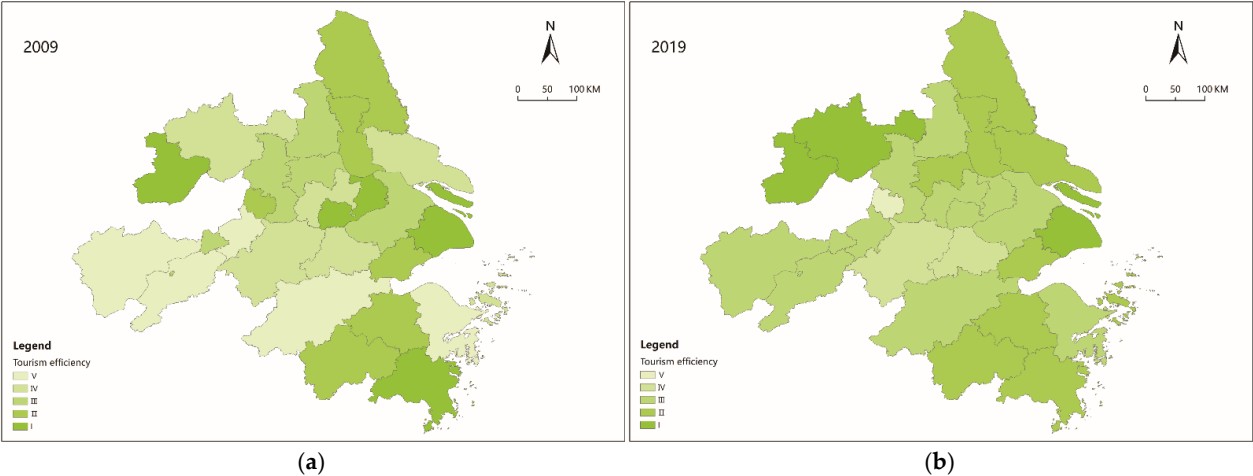

**Figure 1.** Evolution of spatial differences in tourism eco-efficiency in (**a**) 2009 and (**b**) 2019.

Firstly, the mean value of tourism eco-efficiency increased from 0.9788 in 2009 to 1.1167 in 2019, with an average annual growth rate of 1.28%. In general, the mean value of tourism eco-efficiency shows an increasing trend probably because of the implementation of tourism energy conservation and emission reduction projects in these cities, which reduce the impact of the tourism economic activities on the ecological environment and promote the continuous development of the ecological civilization. This conclusion is different from that of Sun et al. Sun et al. believed that during the study period, the tourism eco-efficiency in the Yangtze River Delta showed a fluctuating trend of it first declining, then rising, and then declining [16]. The reason for this is that the index selection of the tourism eco-efficiency measurement that was conducted by Sun et al. was not as comprehensive as it is in this study, and the measurement method is also different from this study.

Secondly, the spatial distribution of tourism eco-efficiency is uneven. The number of cities in ranks II and III contribute to 76.9% of the total number of cities, which indicates that the tourism eco-efficiencies of most of the cities are at a medium-to-high level. In 2019, Shanghai, Chuzhou, and Hefei were classified as grade I cities. Yancheng, Taizhou (Jiangsu), Nantong, Zhenjiang, Shaoxing, Jinhua, Taizhou (Zhejiang), Jiaxing, and Zhoushan were grade II cities. Yangzhou, Nanjing, Changzhou, Wuxi, Suzhou, Anqing, Chizhou, Tongling, Ningbo, Hangzhou, and Wuhu were grade III cities. Huzhou and Xuancheng were grade IV cities. Only Ma'anshan was a grade V city. Again, the tourism eco-efficiency of some cities showed fluctuating trends. Among them, 10 cities showed an increasing trend, including Nantong (IV→II), Zhenjiang (III→II), Changzhou (IV→III), Chuzhou (IV→I),

Anqing (V→III), Chizhou (V→III), Wuhu (V→III), Hangzhou (V→III), Ningbo (V→III), and Zhoushan (IV→II), which are probably due to the rapid growth of the tourism economy, which leads to the rapid improvement of tourism eco-efficiency.

Three cities showed a downward trend, namely Wuxi (I→III), Ma'anshan (I→V), and Wenzhou (I→II). Although these three cities also have a high TRE, the advantages of the tourism resources have not been transformed into advantages for the tourism economy because of the limitations of the unfavorable conditions such as the development of the economy, transportation, location, and science and technology. In addition, the balance between the tourism economic growth and carbon emission and energy consumption has been neglected, which finally leads to the continuous decline of tourism eco-efficiency.

### 3.2. Spatial Correlation Network Analysis of Tourism Eco-Efficiency

#### 3.2.1. Analysis of the Whole Network

Based on the modified gravity model, the spatial correlation matrix of the tourism eco-efficiency of 26 cities was calculated. ArcGIS 10.5 and UCINET 6.0 software were used to construct the spatial correlation network of tourism eco-efficiency in 2009 and 2019 (Figure 2). This is significantly different from the previous spatial analysis based only on the "attribute data" [16]. Based on the "relational data" (spatial correlation matrix), this study further uses the social network analysis to reveal the structural characteristics of the spatial linkages of tourism eco-efficiency network, which is also well applied to the existing research [46]. Based on the analysis of the spatial difference in tourism eco-efficiency in the Yangtze River Delta urban agglomeration, this study combined the analysis of SNA and GIS to construct the spatial correlation network of tourism eco-efficiency in the Yangtze River Delta urban agglomeration. Figure 2 shows that the tourism eco-efficiencies of the 26 cities do not exist in isolation, and they have obvious gravity (correlation) in space. The gravity (correlation) between the cities gradually increased from 2009 to 2019. Previous studies have pointed out that the cities with different tourism eco-efficiency values have the characteristics of cluster and spatial spillover effects [17]. This study is a further exploration of its spatial correlation, and the conclusion is also a validation of its spatial correlation.

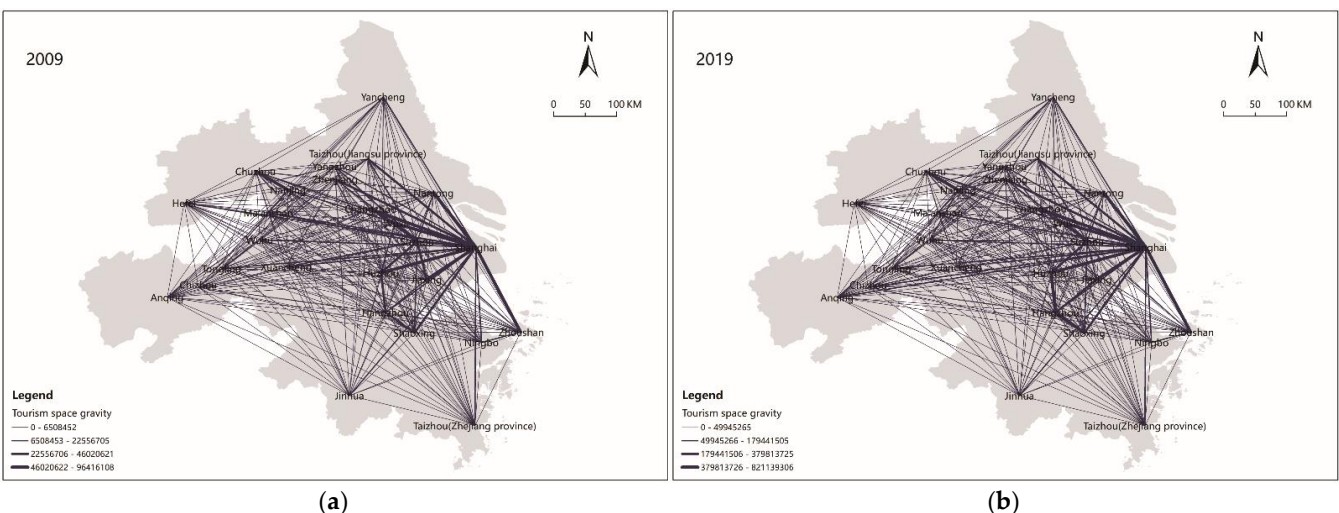

**Figure 2.** Spatial correlation network of tourism eco-efficiency in (**a**) 2009 and (**b**) 2019.

The degree of network correlation was used to measure the closeness of the connections between the different city nodes in the spatial correlation network. From 2009 to 2019, the network connectivity of the spatial correlation network of tourism eco-efficiency in 26 cities was one, indicating that all the cities were connected. The spatial correlation network structure performed well in terms of its connectivity, accessibility, and robustness (Figure 3).

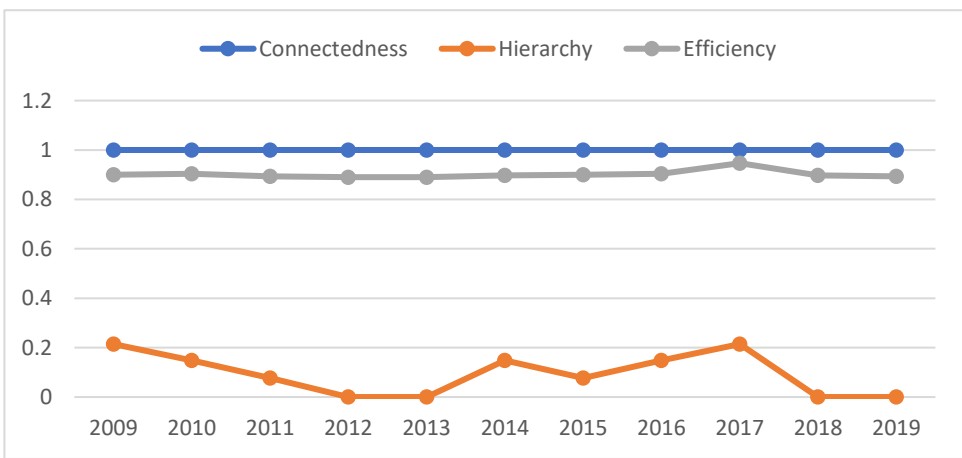

**Figure 3.** Relevance, hierarchy, and efficiency of the tourism eco-efficiency network.

From 2009 to 2019, the hierarchical average of the spatial association network of tourism eco-efficiency was 0.214, which is in the middle and lower levels, and shows a downward trend, indicating that the differences in the tourism eco-efficiencies between the cities are decreasing, and more core nodes of the network have emerged. The network level for 2012, 2013, 2018, and 2019 was 0, indicating that each node has a high possibility of having a spatial spillover effect. On the other hand, the network efficiency was applied to measure the relationships in the spatially connected network. The higher the network efficiency, the more spatial correlations indicated. From 2009 to 2019, the network efficiency of the spatial correlation network was approximately 0.9, which was at a high level, indicating that the network has high correlation and redundancy. These conclusions are consistent with the conclusion of Wang [41].

3.2.2. Analysis of Individual Network Characteristics

UCINET 6.0 software was used to analyze the centrality of the spatial correlation network of tourism eco-efficiency. The degree of centrality is the number of other cities that are directly connected to a city. A high value of the degree of centrality indicates that the city has a high number of connections to other cities. The point-out and point-in degrees characterize the number of relationships that are sent from the city to other cities and the number of relationships received from other cities, respectively. The mean value of the degree of centrality of tourism eco-efficiency was approximately 17. The degree of centrality of Shanghai, Suzhou, Nanjing, and Hangzhou were above the average, indicating that these cities have a high spatial correlation and priority in the tourism eco-efficiency correlation network. The point-out degrees of 26 cities were >0, indicating that the tourism eco-efficiencies of each city have some spatial radiation. Shanghai, Suzhou, Nanjing, and Hangzhou have higher than average point-out degrees, while Yancheng, Taizhou (Zhejiang), Ma'anshan, Anqing, and Chizhou have a smaller number of relationships. In terms of the point-in degree, Shanghai also has the highest number of receiving relationships, up to twenty-four, while Yancheng, Taizhou (Zhejiang Province), and Anqing have the lowest number of receiving relationships with only one.

Closeness centrality considers the average length of the shortest route from each city to the other cities (Table 3). The larger that its value is, then the closer it is to other cities. The closeness centrality of the city ranges from 51 to 100, indicating that each city in the spatial correlation network can be connected with the other cities relatively quickly. The closeness centrality of Shanghai, Nanjing, Suzhou, and Hangzhou were higher than the average value of 54.53, indicating that these cities play the role of "core actors" in the spatial correlation network and they can lead its development, probably because these cities are not only geographically superior, but they are also able to effectively develop and utilize tourism resources such that the economic benefits of tourism are guaranteed. The values of closeness centrality of

22 cities, such as Changzhou, Ningbo, and Wuhu, were lower than the average value of 54.53, indicating that the eco-efficiency of tourism is relatively low, and hence, they are restricted as the other cities make them passive in the spatial correlation network.

**Table 3.** Centrality analysis of the spatial correlation network of tourism eco-efficiency in 2019.

| City | Degree | | | Closeness | Betweenness | City | Degree | | | Degree | Degree |
|------|-----|-----|--------|-----------|-------------|------|-----|-----|--------|--------|--------|
|      | Out | In  | Center |           |             |      | Out | In  | Center |        |        |
| Shanghai | 25 | 24 | 100 | 100.00 | 72.21 | Huzhou | 4 | 4 | 16 | 54.35 | 0.22 |
| Nanjing | 8 | 7 | 32 | 59.52 | 2.56 | Shaoxing | 3 | 3 | 12 | 53.19 | 0.06 |
| Wuxi | 4 | 3 | 16 | 54.35 | 0.11 | Jinhua | 2 | 2 | 8 | 52.08 | 0.00 |
| Changzhou | 2 | 3 | 12 | 53.19 | 0.00 | Zhoushan | 4 | 4 | 16 | 54.35 | 0.39 |
| Suzhou | 13 | 14 | 56 | 69.44 | 11.11 | Taizhou (Zhejiang) | 1 | 1 | 4 | 51.02 | 0.00 |
| Nantong | 2 | 2 | 8 | 52.08 | 0.00 | Hefei | 3 | 2 | 12 | 53.19 | 0.17 |
| Yancheng | 1 | 1 | 4 | 51.02 | 0.00 | Wuhu | 3 | 3 | 12 | 53.19 | 0.05 |
| Yangzhou | 4 | 4 | 16 | 54.35 | 0.05 | Ma'anshan | 1 | 2 | 8 | 52.08 | 0.00 |
| Zhenjiang | 4 | 4 | 16 | 54.35 | 0.05 | Tongling | 2 | 2 | 8 | 52.08 | 0.00 |
| Taizhou (Jiangsu) | 3 | 3 | 12 | 53.19 | 0.05 | Anqing | 1 | 1 | 4 | 51.02 | 0.00 |
| Hangzhou | 7 | 7 | 28 | 58.14 | 1.94 | Chuzhou | 3 | 2 | 12 | 53.19 | 0.05 |
| Ningbo | 2 | 2 | 8 | 52.08 | 0.00 | Chizhou | 1 | 2 | 8 | 52.08 | 0.00 |
| Jiaxing | 3 | 3 | 12 | 53.19 | 0.06 | Xuancheng | 3 | 4 | 16 | 54.35 | 0.27 |

Betweenness centrality reflects the number of bridges that serve as the shortest way to the other two cities (Table 3). The higher the value is, then the more control it has in the network. Shanghai, Nanjing, and Suzhou play a significant mediating role in the spatial correlation network. The sum of the betweenness centrality values of these three cities account for 96% of the total, indicating that they play the role of key nodes in the spatial correlation network, and they can connect various cities and perform well in the transfer process of the tourism production factors and the tourism resources. Similarly, these cities also have a strong ability to control the network.

Shanghai, Suzhou, and Nanjing also have significant characteristics in the aspect of the individual network characteristics. The tourism eco-efficiency of these three cities can be quickly connected with the other cities, and they play the role of "core actors" and play the roles of being a "bridge" and an "intermediary", which is consistent with the conclusion of Wang [41].

*3.3. Block Model Analysis*

In this study, the convergence of the iterated correlations (CONCOR) algorithm in UCINET 6.0 software was used to classify 26 cities into four blocks by selecting two as the maximum segmentation depth and 0.2 as the convergence criterion (Figure 4) [39]. By compared it with the spatial error model (SEM) that has been used by previous scholars to analyze the spatial spillover effect [17], the block model analysis studies the spillover relationship between the blocks, which can more intuitively understand the flow of tourism-related resources between the blocks and the relationships within and between the blocks. The spatial correlation of tourism eco-efficiency of 26 cities in 2019 was further analyzed using the block model. Block I includes Shanghai, Suzhou, and Hangzhou. Block II includes Tongling, Anqing, Hefei, Yancheng, Nanjing, Ningbo, Chizhou, and Taizhou (Zhejiang). Block III includes Huzhou, Changzhou, Xuancheng, Wuxi, Jinhua, Zhoushan, Shaoxing, Jiaxing, and Nantong. Block IV includes Yangzhou, Ma'anshan, Taizhou (Jiangsu), Chuzhou, Zhenjiang, and Wuhu. Table 4 shows 218 spatial network relationships of tourism eco-efficiency in 2019. There are 24 intra-block relationships, accounting for 11% of the total number of relationships. One hundred and ninety-four relationships were between the blocks, accounting for 89% of the total, indicating that the intra-block aggregation effect is relatively weak, while the inter-block spatial correlation is relatively significant. The spatial

spillover effect plays a leading role in the spatial correlation network structure of tourism eco-efficiency. Furthermore, this study compares the internal and external correlations of each block and the ratio of the expected internal relationship to the actual internal relationship. The results show that Shanghai, Suzhou, and Hangzhou in block I are the "core actors" in the spatial correlation network of tourism eco-efficiency probably due to their strong tourism economic strength, obvious location advantages, and high tourism eco-efficiency level which easily attract the inflow of socioeconomic and environmental resources and other factors from other cities. However, due to their relatively backward economic foundation, their remote geographical location, and their low development level of tourism eco-efficiency, the cities in the other three blocks are highly dependent on the cities in block I. Therefore, the spillover effect of block I is very significant.

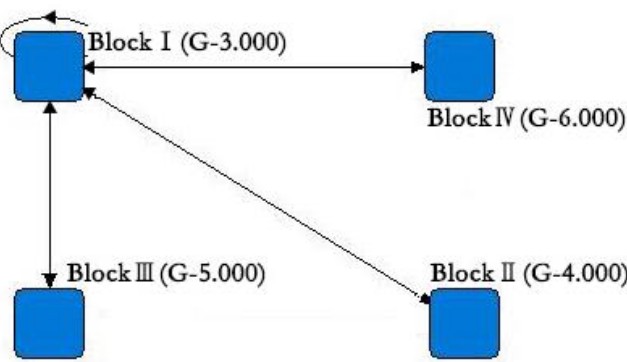

**Figure 4.** Diagram of block model analysis for 2019.

**Table 4.** Spillover effects of tourism eco-efficiency spatially related blocks in 2019.

| | Receiving Relationship Matrix | | | | Number of Members | Number of Receiving Relationships | | Number of Sending Relationships | | Desired Internal Relationship Ratio | Actual Internal Relationship Ratio | Characteristics |
|---|---|---|---|---|---|---|---|---|---|---|---|---|
| | Block I | Block II | Block III | Block IV | | In | Out | In | Out | | | |
| Block I | 4 | 8 | 23 | 10 | 3 | 4 | 41 | 4 | 41 | 8% | 9% | Core actors |
| Block II | 8 | 3 | 1 | 6 | 8 | 3 | 15 | 3 | 16 | 28% | 16% | Marginal actors |
| Block III | 23 | 2 | 3 | 0 | 9 | 3 | 25 | 3 | 24 | 32% | 11% | Marginal actors |
| Block IV | 10 | 6 | 0 | 2 | 6 | 2 | 16 | 2 | 16 | 20% | 11% | Marginal actors |

The network density matrix of the block was calculated based on the results in Table 4. The network density matrix was transformed into the image matrix using some criteria (Table 5). Figure 4 shows the significant centralization trend of the 26 cities, and the block model structure in 2019 is a typical "core-edge structure". All of the relationships in the spatial correlation network of tourism eco-efficiency are from block I. Only block I has a value of one on the diagonal of the homogeneous matrix, indicating that block I has a reflexive relationship, that is that it can supply to other cities well, while satisfying itself, and its internal tourism eco-efficiency has a more significant correlation. There is a bidirectional spillover relationship between block I and the other three blocks; however, the spillover relationships among blocks II, III, and IV are not significant.

In terms of the overall spatial distribution pattern, the spillover of block I, which is the core of the spatial correlation network, is the most obvious. Block III has the second highest number of spillover relationships, which mainly spillover to block I. Blocks II and IV have the same number of spillover relationships. Blocks II and IV are at the outermost parts of the entire network, confirming that the average network density in the whole network is at a moderate-to-low level, and the network is relatively loose. This also confirms the "low network hierarchy" conclusion that is mentioned above, where the members of the blocks are only classified into "core actors" and "fringe actors". The spatial variation of tourism eco-efficiency combined with the conclusion of this study indicates that the tourism eco-efficiencies of Shanghai, Suzhou, and Hangzhou are more fragmented, and this narrows the

gap in the tourism eco-efficiency within the region, and the Matthew effect is sufficiently mitigated to be more beneficial to the tourism eco-efficiency of the overall network.

**Table 5.** Density and image matrixes of spatially related blocks of tourism eco-efficiency.

| | **Density Matrix** | | | | | **Image Matrix** | | | |
|---|---|---|---|---|---|---|---|---|---|
| | **Block I** | **Block II** | **Block III** | **Block IV** | | **Block I** | **Block II** | **Block III** | **Block IV** |
| Block I | 0.667 | 0.333 | 0.852 | 0.556 | Block I | 1 | 1 | 1 | 1 |
| Block II | 0.333 | 0.054 | 0.028 | 0.125 | Block II | 1 | 0 | 0 | 0 |
| Block III | 0.852 | 0.014 | 0.042 | 0.000 | Block III | 1 | 0 | 0 | 0 |
| Block IV | 0.556 | 0.125 | 0.000 | 0.067 | Block IV | 1 | 0 | 0 | 0 |

*3.4. Influencing Factors*

3.4.1. Selection of Influencing Factors

The occurrence and development of the spatial correlation network of tourism eco-efficiency is a direct reflection of the interaction and cooperation of the resource elements in the tourism geographical spaces of 26 cities. The resultant force formed by different influencing factors in the interrelation leads to the realization of this process. The change in resultant force strength [40] affects the degree of development of the structure of the spatial correlation network.

The EDL directly affects the cooperation of tourism projects between the cities, and then, affects the development level of the spatial correlation network structure of tourism eco-efficiency. Therefore, gross domestic product (GDP) was used in this study to represent the level of regional economic development [41].

The status of the tourism industry can reflect the government's emphasis on regional tourism development and the ability for the regional tourism factors to agglomerate. The higher the development level of the tourism industry is, then the wider the flow space of the regional tourism production factors is. Therefore, this study selected the total income from tourism as a proportion of the GDP to represent the status of the tourism industry [47].

Capital radiates and diffuses into the frontier regions through various media, which improve the quality and efficiency of regional tourism, and it is measured by the investment in fixed assets for tourism [40]. The TRE influences the flow and configuration of the tourism input factors among the provinces and municipalities, thus influencing the division of labor and cooperation of their tourism industries as determined by the total number of scenic spots at different levels [48].

Gravity is inversely proportional to the square of the distance, and the path dependence can be regarded as the inertia. The closer the distance between the two cities is, then the more favorable the spillover of tourism eco-efficiency is. The spillover between the cities is more likely to follow the existing path (inertia), which is represented by the spherical DBC [49]. The IDL is an important pathway of spatial agglomeration and the radiation of tourism economy between the cities, and it is expressed in terms of total postal and telecommunications services [50].

In this study, the spatial correlation matrix of tourism eco-efficiency of 26 cities in 2019 was selected to be the explained variable. The difference matrices of the EDL, the status of the tourism industry, the TIL, the TRE, the DBC, and the IDL were used as the explanatory variables in the QAP analysis (Table 6).

**Table 6.** Influencing factors for the construction of the index of the spatial correlation network of tourism eco-efficiency.

| Indicator Name | Primary | Expectation | References |
|---|---|---|---|
| Economic development level | Gross Domestic Product | + | Liu et al. [22]; Wang et al. [51]. |
| The status of the tourism industry | Total income from tourism accounts for the proportion of GDP | + | Liu et al. [22]; Hu et al. [35]; Wang et al. [51]. |
| Tourism investment level | Investment in fixed assets for tourism | + | Wang et al. [38]; Wei et al. [40]. |
| Tourism resources endowment | Total number of A-level scenic spots (spots) | + | Hu et al. [35]; Lu et al. [48]. |
| Distance between cities | Spherical distance between cities | − | Wang et al. [38]; Liu [49]. |
| Information development level | Total postal and telecommunications services | + | Wang et al. [38]; Chen [50]. |

### 3.4.2. Analysis of Influencing Factors

The driving factors have an important influence on the formation and evolution of the spatial correlation network of tourism eco-efficiency. Therefore, the QAP analysis is used to measure the influence of various driving factors on the nature, direction, and intensity of the spatial correlation of tourism eco-efficiency. The QAP is a method based on re-sampling, which can successfully eliminate the false structural relationship. This method has been very common in the research of tourism, sociology, and geography [38,39]. The QAP correlation and regression analyses were performed to outline the influencing factors of the spatial correlation network of tourism eco-efficiency in 26 cities. UCINET 6.0 software was used to conduct the QAP correlation analysis on the spatial correlation matrix and influencing factor matrix of tourism eco-efficiency. Five thousand random permutations were selected, and the correlation analysis results are shown in Table 7.

**Table 7.** Results of QAP correlation analysis.

| Independent Variable | QAP Correlation Analysis | |
|---|---|---|
| | Correlation Index | *p* Value |
| Economic development level | 0.585 *** | 0.000 |
| Status of the tourism industry | −0.039 | 0.398 |
| Tourism investment level | −0.145 ** | 0.032 |
| Tourism resources endowment | 0.112 | 0.127 |
| Distance between cities | −0.233 *** | 0.000 |
| Information development level | 0.620 *** | 0.000 |

Note: *** and ** indicate significance levels of 0.001 and 0.05, respectively.

The difference matrices of the EDL and the IDL are positively correlated with the spatial correlation network of tourism eco-efficiency. The correlation coefficient is positive and significant at the 1% level, which confirms the research hypothesis. This further indicates that the higher the EDL and IDL are, then the more conducive they are for to the formation and development of the spatial correlation network of tourism eco-efficiency.

The difference matrix of the DBC is negatively correlated with the spatial correlation network of tourism eco-efficiency. The correlation coefficient is positive and significant at the 1% level, which indicates that the closer the DBC is, then the more conducive it is for the formation and development of the spatial correlation network of tourism eco-efficiency. The cities with higher tourism eco-efficiencies will preferentially affect nearby the cities with lower tourism eco-efficiencies, and then, they will gradually promote the improvement of tourism eco-efficiency of the whole region.

The difference matrix of the TIL is negatively correlated with the development of the spatial correlation network of tourism eco-efficiency. The correlation coefficient is negative and significant at the 5% level, which is not consistent with the research hypothesis, indicating that tourism capital hinders the improvement of inter-city tourism eco-efficiency.

The correlation between the difference matrices of the TIS and the TRE and the spatial correlation network of tourism eco-efficiency were not strong, and both of them were not significant probably because the status of the tourism industry reflects the government's intention and goal to achieve regional economic development based on tourism. The TISs of the different cities are different, which is not conducive for the cooperation between the cities with low and high TIS values, and it further hinders the formation and development of the spatial correlation network of tourism eco-efficiency. Tourism resources are essential for the development of the tourism industry. Because of the similarity of the tourism resources in various cities, the competition among the cities is fierce, which hinders the formation and development of the spatial correlation network of tourism eco-efficiency.

Based on the correlation analysis, this study selected the matrices of difference of the EDL, the TIL, the DBC, and the IDL for the QAP regression analysis. The results of the QAP regression analysis are shown in Table 8. The adjusted $R^2 = 0.487$, which is significant at the 1% level, indicates that the four influencing factor matrices explain 48.7% of the spatial correlation of tourism eco-efficiency.

**Table 8.** Results of QAP regression analysis.

| Independent Variable | QAP Regression Analysis | | |
|:---:|:---:|:---:|:---:|
| | Unstandardized Coefficients | Standardized Coefficients | *p* Value |
| Economic development level | 0.2537 *** | 0.3241 *** | 0.0005 |
| Tourism investment level | 0.0123 | 0.0159 | 0.3408 |
| Distance between cities | −0.1626 *** | −0.2176 *** | 0.0005 |
| Information development level | 0.3681 *** | 0.4072 *** | 0.0005 |
| Intercept | 0.0748 | 0.0000 | 0.0000 |
| | $R^2 = 0.4900$ *** | Adjusted $R^2 = 0.4870$ *** | 0.0000 |

Note: *** indicate significance at the levels of 0.001.

The regression coefficient of the relationship matrix between the EDL and the urban informatization development level is significantly positive at the 1% level, indicating that the high EDL promotes the improvement of the linkage intensity of tourism eco-efficiency. The stronger the economy is, then there will be better circulation of the tourism production resources and better linkages between the tourism eco-efficiencies among the cities. This conclusion is consistent with the study of Liu et al. [22,52–54]. Meanwhile, the higher the IDL is, then the more convenient and efficient the circulation of the tourism production factors and the transfer of materials and information will be, which improve tourism eco-efficiency. This conclusion is consistent with the findings of Wang et al. [18,55].

The DBC is significantly negative at the 1% level, indicating that a more distant geographical location is essential to strengthen the correlation between the tourism development efficiencies among the cities, and the more distant the geographical distance between the efficient and inefficient cities is, then the less conducive it is for the transfer and circulation of production factors and information, thus hindering the improvement of the correlation intensity of tourism eco-efficiency. This finding is in line with that of Cheng et al. [56].

The difference matrix of the TIL is not significant, indicating that tourism capital insignificantly hinders the inter-city tourism production cooperation. The capital is usually concentrated in the economically developed cities, which increases the gap between the cities with high and low tourism eco-efficiencies; however, whether it hinders the development of the spatial correlation network of tourism eco-efficiency is not obvious.

## 4. Conclusions

### 4.1. Findings

This study established an index system for the measurement of tourism eco-efficiency. The values of tourism eco-efficiency of 26 cities in the Yangtze River Delta from 2009 to 2019 were measured using the undesirable output super-SBM model. In addition, the spatial

correlation matrix of tourism eco-efficiency was constructed using the modified gravity model. The spatial correlation of tourism eco-efficiency and its influencing factors were analyzed using the social network analysis.

Firstly, our study differs from previous studies that simply measured the tourism eco-efficiency of a tourism destination and analyzed their spatiotemporal differences [16]. This study argues that the tourism eco-efficiencies of different regions have spatial correlations and are not independent of each other. Based on the analysis of the spatial differences of the tourism eco-efficiencies of the cities in the Yangtze River Delta, this study combined the social network and the geographic information system analyses to construct a spatial correlation network of tourism eco-efficiency, which facilitates the extension of the original "point measurement" of tourism eco-efficiency to the level of "surface evaluation", and also allows for a comprehensive and in-depth exploration of its internal correlations to be made, thus enabling more insightful conclusions to be drawn. Strategically, this combined social network analysis and geographic information system approach can facilitate research on the formation, evaluation, and evolution of the spatially linked networks of tourism eco-efficiency.

Secondly, the driving factors for the formation of the spatial association network of tourism eco-efficiency are analyzed to expand the scope of research. At present, scholars mainly focus on the influencing factors of the spatiotemporal evolution of tourism eco-efficiency [8,51], and much less research has been conducted on the driving factors of the formation of the spatial association network of tourism eco-efficiency. Therefore, the QAP analysis was applied to measure the influence of various driving factors on the nature, direction, and intensity of the spatial linkages of tourism eco-efficiency. The results show that the EDL, the DBC, and the IDL are key factors that have significant effects on the spatial correlation network of tourism eco-efficiency in 26 cities in the Yangtze River Delta. In addition, the nature, direction, and degree of influence of various driving factors vary.

The empirical results verify that the tourism eco-efficiencies of 26 cities are spatially correlated with spatial differentiation patterns and dynamic development trends. On the one hand, tourism eco-efficiency shows an overall growth trend, indicating that with the formulation and implementation of tourism energy conservation and reduction projects in various cities, the construction of the ecological civilization has been continuously promoted. Therefore, the impact of the tourism economic activities on the ecological environment is reduced thereby promoting the maximization of social and economic benefits. On the other hand, the spatial distribution of tourism eco-efficiency in the Yangtze River Delta is not uniform. The tourism eco-efficiencies of most of the cities are at a medium-to-high level, and some cities show a fluctuating change trend. Ten cities show an increasing trend, and three cities show a decreasing trend.

This study also analyzed the degrees of correlation, the network levels, and the network efficiency values of the spatial correlation networks of tourism eco-efficiency in 2009 and 2019 to effectively judge their scale and evolution in terms of achieving tourism eco-efficiency [38,39]. By comparing the correlation degrees of the spatial correlation network of tourism eco-efficiency in 2009 and 2019, this study finds that all of the cities are interrelated, and there is no "island" phenomenon. The structure of the spatial correlation network of tourism eco-efficiency shows good connectivity, accessibility, and robustness. The network rank is in the lower middle level, and it shows a decreasing trend, indicating that the differences between the cities are shrinking and more core nodes are emerging. Moreover, the network efficiency is at a high level, indicating that there are more correlations and only some redundant relationships in this network.

This study further analyzed the structure of the nodes in the spatial correlation network of tourism eco-efficiency in 2009 and 2019. The degrees of centrality index of the nodes in the tourism eco-efficiency association network were measured to effectively judge their roles, spatial differences, and evolution [38,39]. Shanghai, Suzhou, Nanjing, and Hangzhou had above average degrees of centrality, indicating that these cities have more connections, and they are a priority in the spatial correlation network of tourism eco-efficiency. The point-out

degrees of 26 cities were >0, indicating that the tourism eco-efficiency of each city has some spatial radiation. Shanghai, Suzhou, Nanjing, and Hangzhou had higher than average point-out degrees. Yancheng, Taizhou (Zhejiang), Ma'anshan, Anqing, and Chizhou had lower point-out degrees. In terms of the point-in degree, Shanghai had the highest number of receiving relations, up to twenty-four, while Yancheng, Taizhou (Jiangsu) and Anqing had the lowest number of receiving relations of only one. In addition, the tourism eco-efficiency of each city in the spatial association network can produce connections with other cities relatively quickly. Shanghai, Nanjing, Suzhou, and Hangzhou had higher than average centrality values, indicating that they play the role of "core actors" in the spatial correlation network of tourism eco-efficiency, leading to the development of the whole network. The tourism eco-efficiencies of 22 cities, such as Changzhou, Ningbo, and Wuhu, were lower than the average level, indicating that their eco-efficiency of tourism is relatively low, leading to the restrictions of other cities and passive positions in the network. Shanghai, Suzhou, and Nanjing play a significant role as "bridges" and "intermediaries", indicating that they play a key role in the spatial association network of tourism eco-efficiency by connecting well with other cities in the transfer of tourism production factors, and they have a strong control in the network.

The spatial correlation of tourism eco-efficiency of 26 cities in 2019 was further analyzed using the block model. Block I is the "core actor" of the spatial correlation network of tourism eco-efficiency with the maximum spillover relationships, and it presents a "two-point" distribution in space. Block III has the second highest number of spillover relationships, and it is centered around the spatially central actor, and it mostly spills over to block I. Blocks II and IV have the same number of spillover relationships and are located in the outermost layer of the entire network. The mutual spillover relationships among blocks II, III, and IV are fewer, and therefore they are "marginal actors" in the spatial correlation network of tourism eco-efficiency.

The tourism eco-efficiency levels of Shanghai, Hangzhou, and Suzhou are relatively high because of their developed tourism economy and geographical advantages. It is easy for these cities to attract social, economic, and environmental resources from other cities. However, the cities in the other three blocks have a relatively backward economic foundation and a poor geographical location, and their tourism eco-efficiency development level is also relatively low with a high dependence on block I through the spillover effect.

All of the relationships in the spatial correlation network of tourism eco-efficiency originate from block I. In other words, Shanghai, Suzhou, and Hangzhou play driving roles in improving the tourism eco-efficiency of the other cities significantly, thereby narrowing the gap in the Yangtze River Delta region and fully alleviating the Matthew effect.

The formation and development of the spatial correlation network of tourism eco-efficiency are influenced by many factors. The results of the QAP correlation analysis show that the EDL and the IDL and the DBC and the TIL are positively and negatively correlated, respectively, with the development of the spatial correlation network of tourism eco-efficiency. The QAP regression analysis shows that the EDL, the DBC, and the IDL play significant roles in the spatial correlation network of tourism eco-efficiency. Therefore, the improvement of tourism eco-efficiency can be achieved by improving the EDL and the IDL. Meanwhile, the closer the cities are to the high eco-efficiency cities, the more beneficial it is to improve their tourism eco-efficiencies. The TIS, the TIL, and the TRE had no significant relationship with the spatial correlation network of tourism eco-efficiency.

The methods and findings of this research can be useful in enhancing the tourism eco-efficiency in the Yangtze River Delta region of China, and they can be applied to multiple destinations. Further empirical studies by other scholars are required to verify the outcomes of this research.

*4.2. Implications*

Based on the established tourism eco-efficiency measurement index system, this study used the undesirable output Super-SBM model to measure the tourism eco-efficiency values

of 26 cities in the Yangtze River Delta in 2009 and 2019. Then, the gravity model was used to calculate the spatial correlation of tourism eco-efficiency of 26 cities in the Yangtze River Delta, and the relationship matrix was built. Finally, the social network analysis was used to analyze the spatial correlation of tourism eco-efficiency and its influencing factors. This study is a bold attempt, which not only enriches the research content of tourism eco-efficiency, but it also promotes the sustainable development of regional tourism.

### 4.2.1. Theoretical Implications

Firstly, this study is in line with China's development goal of achieving an ecological civilization construction, as well as its requirements for high-quality economic development. It is of great significance to enrich the theory of sustainable development and expand the application scope of the sustainable development theory.

Secondly, this study is a bold application of the theory of tourism eco-efficiency. By exploring the tourism eco-efficiency of different cities in the Yangtze River Delta, this study further compares the spatial and temporal differences, the spatial correlation, and the influencing factors. This is the logical starting point for exploring the green and sustainable development of the Yangtze River Delta urban agglomeration. This study is different from previous studies that simply measure the tourism eco-efficiency of a certain tourist destination and analyze its internal spatio-temporal differences. It believes that the tourism eco-efficiency of different regions is not independent of each other, but they have a certain spatial correlation. The overall network structure characteristics of the spatial correlation of tourism eco-efficiency were analyzed from the perspective of their "relationship".

Thirdly, on the basis of analyzing the spatial differences of tourism eco-efficiency in different cities in the Yangtze River Delta, this study combined the analysis of the SNA and GIS data to build the spatial correlation network of tourism eco-efficiency in the Yangtze River Delta urban agglomeration. This is not only conducive to further promoting the original "point measurement" of tourism eco-efficiency to a "surface evaluation", but also to comprehensively and deeply explore its internal correlation, so as to draw more insightful conclusions. Strategically, the combination of SNA and GIS can promote the research on the formation, evaluation, and evolution of the spatial correlation network of tourism eco-efficiency.

### 4.2.2. Management Implications

This study found that the overall level of tourism eco-efficiency in the Yangtze River Delta region is constantly improving, and the tourism eco-efficiencies of most of the cities in the region are also improving. Through the method of the social network analysis, the network characteristics between the cities in the region were further analyzed. This can not only help the government to better coordinate the relationship between the regional economic development and the ecological environment and formulate policies according to the local conditions, but also provide new ideas for the urban development of ecological tourism.

Based on the block model analysis, one core block and three marginal blocks were distinguished in this study. It is helpful for the local government to take into account the characteristics of the different sectors and adopt corresponding management strategies to narrow the gap of eco-efficiency of urban tourism in the region to alleviate the Matthew effect.

In addition, this study also explores the influencing factors of the spatial correlation network of tourism eco-efficiency, which can provide a reference for the government and tourism enterprises to formulate the formation and optimization strategies of the spatial correlation network of tourism eco-efficiency.

### 4.3. Limitations

This study has some limitations. Firstly, the input–output mismatch problem is prone to occur from the perspective of the multi-factor evaluation [8]. Secondly, this study was unable to obtain the energy input and consumption data for the tourism-related aspects such as aviation, catering, shopping, and entertainment. In addition, air pollution should

include also No$_x$ and particulate matter concentration data (e.g., PM5, PM2.5, PM1.0) [57,58]. However, due to the limited number of data, the concentration of nitrogen oxides and particulate matter in the study area was very small, so this aspect is not reflected in the established index system. Thirdly, tourism eco-efficiency evaluation should theoretically cover three dimensions: economy, ecology, and society, including the satisfaction of the local residents and tourists as indicators. Therefore, the evaluation system for tourism eco-efficiency should be further improved, theoretically, to provide reference values. These improvements are necessary to provide a strong support for the design of appropriate policies and appropriate decision making.

**Author Contributions:** Conceptualization, Y.W. and H.C.; methodology, L.A.; software, L.A. and Y.Z.; validation, Y.W.; formal analysis, Y.W. and H.C.; resources, Y.W. and H.C.; data curation, L.A.; writing—original draft preparation, Y.W. and H.C.; writing—review and editing, Y.W. and H.C.; visualization, L.A. and Y.Z.; supervision, Y.W. and H.C.; funding acquisition, Y.W. and H.C. All authors have read and agreed to the published version of the manuscript.

**Funding:** This research study was supported by the National Social Science Foundation of China (to Yuewei Wang) (Grant No. 19BGL145).

**Institutional Review Board Statement:** Not applicable.

**Informed Consent Statement:** Not applicable.

**Data Availability Statement:** Not applicable.

**Acknowledgments:** The authors would like to acknowledge all experts' contributions in the building of the model and the formulation of the strategies in this study. All individuals included in this section have consented to the acknowledgment.

**Conflicts of Interest:** The authors declare no conflict of interest.

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
