# Peer review of "Spatial Correlation and Influencing Factors of Tourism Eco-Efficiency in the Urban Agglomeration of the Yangtze River Delta Based on Social Network Analysis"

_land, doi:10.3390/land11112089_

Round 1

Reviewer 1 Report

The authors put their efforts in order to develop a new model. The relevant literature is consulted, and this research is based on the inputs of previous researchers. The authors are aware of objective limitations of this paper and of the model presented in the paper and they do not hesitate to express those limitations.

The paper is relevant platform for further research.

Reviewer 2 Report

I think it is a well written paper with an interesting and innovative application of Social Network Analysis.  I like it! Anyway I suggest to enrich the paper by reading these three contributions: 1) Scott, N.; Baggio, R.; Cooper, C. Network Analysis and Tourism: From Theory to Practice; Channel View
Publications Clevedon: Somerset, UK, 1998.
2) Birendra, K.C.; Morais, D.B.; Peterson, M.; Seekamp, E.; Smith, J.W. Social network analysis of wildlife
tourism microentrepreneurial network. Tour. Hosp. Res. 2019, 19, 158–169. 3) Cooper, C.; Scott, N.; Baggio, R. Network Position and Perceptions of Destination Stakeholder Importance. Anatolia 2009, 20, 33–45.

Moreover, "Conclusions" must be enriched with policy implications.

Reviewer 3 Report

Line 11: Please rephrase the first sentence. “Tourism eco-efficiency is significant for….”. Please try to demonstrate the purpose of the study and it’s position in the current scientific literature in this field. The first sentence needs to be more specific in this sence.

Please keep the sentences in neutral form. They can’t start with “We constructed…”, but should be formulated as “The tourism eco-efficiency evaluation index was constructed in the study and used…..”. Please explain separately the in

Please follow the following sequence in constructing the abstract: Purpose. Design. Methodology. Approach, Findings. Research limitations. Research implications. Practical implications. Social implications. Originality.

The whole logical construction in which the research is based in both faulty in many aspects as well as ill presented. This needs to be improved. For example, the research question 1 deals both with spatial as well as temporal aspects which is too much for one research question. Than 2 and 4 appear to be repeating the same question but with focus on temporal (2.) and spatial (4.). The discussion section needs to be structure according to the research questions and provide concrete answers to each of the posed research questions.

The first sentence in the abstract and the first paragraph in the introduction need to answer the question what is the goal/purpose of the study? Out of this goal, 3-5 research questions need to be defined, which clearly lead to achieving the set goal of the study. Why is it important to know the spatial correlation of tourism eco efficiency? This is not an easy question to answer and it should be well thought out.

Additional literature which is relevant and hasn’t been considered in the literature review nor discussed in light of the present results:

Qiu, X., Fang, Y., Yang, X., & Zhu, F. (2017). Tourism eco-efficiency measurement, characteristics, and its influence factors in China. Sustainability, 9(9), 1634.

Peeters, P., Gössling, S., Ceron, J. P., Dubois, G., Patterson, T., Richardson, R. B., & Studies, E. (2017). The eco-efficiency of tourism. Advances in tourism climatology, 105.

Wang, R., Xia, B., Dong, S., Li, Y., Li, Z., Ba, D., & Zhang, W. (2020). Research on the spatial differentiation and driving forces of eco-efficiency of regional tourism in China. Sustainability, 13(1), 280.

Zhang, F., Yang, X., Wu, J., Ma, D., Xiao, Y., Gong, G., & Zhang, J. (2022). How New Urbanization Affects Tourism Eco-Efficiency in China: An Analysis Considering the Undesired Outputs. Sustainability, 14(17), 10820.

An, C., Muhtar, P., & Xiao, Z. (2022). Spatiotemporal Evolution of Tourism Eco-Efficiency in Major Tourist Cities in China. Sustainability, 14(20), 13158.

Li, S., Ren, T., Jia, B., & Zhong, Y. (2022). The Spatial Pattern and Spillover Effect of the Eco-Efficiency of Regional Tourism from the Perspective of Green Development: An Empirical Study in China. Forests, 13(8), 1324.

The discussion whould answer the following questions both in relation to the present as well as the suggested new literature:

How the present results compare to the previous studies in terms of methodology deployed as well as results obtained?

What are the novel or different aspects of the deployed methodology and the obtained results?

How is the methodology used already previously been deployed and validated and how do the obtained results  confirm the findings of the previous studies?

All three questions should cite previous sources for reference.

Another important issue with the indicators system itself is it’s concentration on wastewater, SO2 and smoke emissions as only three undesirable outputs, which is rather narrow in my opinion. Air pollution should include also Nox and particulate matter concentration (e.g. PM5, PM2.5, PM1.0). See for example: United Nations World Tourism Organization. (2004). Indicators of Sustainable Development for Tourism Destinations. Madrid: UNWTO.

Social network analysis is mentioned for the first time in the findings, but no reference do this method was made beforehand in the methodology section. Please carefully explain precisely what type of social network analysis was used, what were the data sources.

EDL, TIL, DBC and IDL, TIS, TRE need to be defined the first time they appear in the text.

Reviewer 4 Report

Recent bibliographic references are limited.
Except for some more recent works by Chinese authors, bibliographic references,  such as citations by recognized specialists in the field, are few in number and not recent.

The works considered relevant are not taken into account, especially those from the main flow of publications (Clarivate Analytics Web of Science or other valuable international and national indexed data bases).

The authors  did not indicate how the results are in relation to previous  expectations and research.

The statements in the conclusions are not supported by the results.

The authors do not explain how research is a step forward for scientific knowledge.

Round 2

Reviewer 3 Report

  Dear authors,

thank you for providing the rsponse to the reviews. Please see below some further changes needed in order to have consistently neutral form in the abstract as well as regarding your reposnse which should be included in the limitations of the study.

Please inspect the whole abstract for personal forms and change them to neutral forms. For example “We constructed the…” on the line 16 should be formulated into “ Tourism eco-efficiency evaluation index system and the undesirable output super-slacks-based measure model were constructed  and deployed….”. This repeats at line 18. The document is now fine, but the abstract still needs to be updated.

2.       The point 8 and the response  (shown below) should be integrated into limitations of the study, and the sources should be cited:

----

Point 8: Another important issue with the indicators system itself is it’s concentration on wastewater, SO2 and smoke emissions as only three undesirable outputs, which is rather narrow in my opinion. Air pollution should include also Nox and particulate matter concentration (e.g. PM5, PM2.5, PM1.0). See for example: United Nations World Tourism Organization. (2004). Indicators of Sustainable Development for Tourism Destinations. Madrid: UNWTO.

Response 8: Due to the limited availability of data, there are few data on nitrogen oxide and particulate matter concentrations in the study area. In other studies, some papers also refer to this research index system, such as Yang, Y. Yan, J.; et al. The spatio-temporal evolution and spatial spillover effect of tourism eco-efficiency in the Yellow River Basin: based on data from the 73 cities. Acta Ecologica Sinica, 2022(20):1-11. Therefore, this paper establishes this index system on the basis of previous research.

  ----

Good luck with the changes!

Reviewer 4 Report

Considering the above, I recommend the authors to consider in the future articles the papers considered relevant, especially those in the main flow of publications 
